# Energy Trading among Power Grid and Renewable Energy Sources: A Dynamic Pricing and Demand Scheme for Profit Maximization

**DOI:** 10.3390/s21175819

**Published:** 2021-08-30

**Authors:** Yoon-Sik Yoo, Seung Hyun Jeon, S. H. Shah Newaz, Il-Woo Lee, Jun Kyun Choi

**Affiliations:** 1Intelligent Convergence Research Laboratory, Electronics and Telecommunications Research Institute (ETRI), Daejeon 34129, Korea; midasyoo@etri.re.kr (Y.-S.Y.); ilwoo@etri.re.kr (I.-W.L.); 2School of Electrical Engineering, Korea Advanced Institute of Science and Technology (KAIST), Daejeon 34141, Korea; jkchoi59@kaist.edu; 3School of Computing and Informatics, Universiti Teknologi Brunei (UTB), Jalan Tungku Link, Gadong BE1410, Brunei; shah.newaz@utb.edu.bn; 4KAIST Institute for Information Technology Convergence, Korea Advanced Institute of Science and Technology (KAIST), Daejeon 34141, Korea

**Keywords:** demand, dynamic pricing, renewable energy certificate (REC), dual decomposition, optimization, energy broker, energy storage system (ESS), distributed energy resources (DERs), smart grid

## Abstract

With the technical growth and the reduction of deployment cost for distributed energy resources (DERs), such as solar photovoltaic (PV), energy trading has been recently encouraged to energy consumers, which can sell energy from their own energy storage system (ESS). Meanwhile, due to the unprecedented rise of greenhouse gas (GHG) emissions, some countries (e.g., Republic of Korea and India) have mandated using a renewable energy certificate (REC) in energy trading markets. In this paper, we propose an energy broker model to boost energy trading between the existing power grid and energy consumers. In particular, to maximize the profits of energy consumers and the energy provider, the proposed energy broker is in charge of deciding the optimal demand and dynamic price of energy in an REC-based energy trading market. In this solution, the smart agents (e.g., IoT intelligent devices) of consumers exchange energy trading associated information, including the amount of energy generation, price and REC. For deciding the optimal demand and dynamic pricing, we formulate convex optimization problems using dual decomposition. Through a numerical simulation analysis, we compare the performance of the proposed dynamic pricing strategy with the conventional pricing strategies. Results show that the proposed dynamic pricing and demand control strategies can encourage energy trading by allowing RECs trading of the conventional power grid.

## 1. Introduction

With technological advancement and growing awareness on how fossil-fuel-based energy generation is contributing unprecedentedly to global warming and climate change, more and more people are adopting eco-friendly distributed energy resources (DERs) (e.g., harnessing the Sun’s energy using solar photovoltaic cells). Despite the fact that DERs producing energy from renewable energy sources (RESs) have weather-dependent output, they are increasingly becoming common in different parts of the world (they are growing both in numbers and production capacity) [1,2,3]. Generally, the energy produced from DERs is distributed within a cluster, and the energy exchange market is in charge of facilitating the energy trading. However, as the consumer-owned DERs are increasing in numbers and so does their production, direct and distributed energy trading is required not only among consumers but also among the utility companies and consumers [3,4]. Therefore, this brings about the necessity of an energy broker, which allows to trade energy among DERs [4]. In order to manage energy use and trade efficiently, the IoT-based smart agents [5,6] can facilitate communication between these DERs and the energy broker, thereby allowing sharing information (e.g., energy generation and consumption information). In addition, when these smart agents are applied, new DERs can be added to the consumer side with the existing deployed DERs without modification of operation (e.g., grid connected mode/isolated mode) [7]. Unlike the existing independent system operator (ISO) [8], an energy broker is equipped with energy storage systems (ESSs), and it can purchase electricity from the sellers and sell directly to the buyers. It can even store the purchased energy in its ESS(s) and sell later. Note that the ISOs may form wholesale power markets where the power transmission operators, utilities and power consumers can trade.

Due to these energy trading shifts from the conventional energy exchange to energy brokers, dynamic pricing is considered for balancing energy demand and distribution [9]. At a power grid, based on monopolistic energy trading, fixed pricing is effective for forecasting demand and deciding on the required amount of electricity generation. However, when it comes to trading energy in a DER-based smart grid, the dynamic pricing is more effective in balancing consumers’ demand as well as reducing the cost of energy distribution. To encourage DER owners to produce energy from RESs, a number of effective measures have been taken. To verify whether the energy is produced from an RES, an energy broker may invoke a DER to provide a renewable energy certificate (REC) [10,11]. RECs are US marketable, intangible energy products that demonstrate the ability to generate and transmit 1MWh of electricity from suitable RESs to the consumers through the power distribution network. Therefore, the DERs that do not provide RECs may not be allowed to trade energy. Furthermore, RECs are a tradable commodity, and they can be used for energy trading. In Europe, to differentiate consumption between RESs and the conventional energy, the Guarantees of Origin (GO) certificate has been introduced [12]. Similarly, in the USA, Republic of Korea, and India, the government and utility companies are encouraging energy providers to use RECs when taking part in energy trading. Undoubtedly, incorporating REC in energy trading can bolster the investment in deploying RES-based DERs so as to reduce the carbon footprint for energy production. Therefore, a dynamic pricing policy and demand control need to be investigated taking into account DER energy generation and energy providers’ RECs trading in an energy broker-orientated trading market.

Previous research findings in [13,14,15,16,17,18,19] impart that a dynamic pricing approach has a strong influence on energy providers’ profit and consumers’ energy demand. According to the proposal in [13], a group of users opts for buying energy at a fixed price, while the other group chooses dynamic pricing. The authors in [13] consider an energy provider’s revenue and demand in their proposal. However, according to this proposal, the users are not equipped with DERs and energy storage facilities. Furthermore, in their solution, the authors assume that a user acts only as a consumer (i.e., a user does not become part of an energy trading). In [14], the authors propose a novel dynamic pricing scheme in order to provide users with a price certainty as well as guaranteed uninterrupted power supply. The authors demonstrate how historical market data can contribute to making their scheme successful in energy sharing. Perhaps a serious disadvantage of this scheme is that it does not consider a broker model. In [15], the authors aim at increasing the availability of renewable energy. In order to attain this goal, they introduce a contract-based adaptive pricing mechanism based on RES distribution. Additionally, in their solution, users are equipped with ESSs in order to reduce peak energy demand. However, this contribution in [15] would have been far more convincing if the authors had considered operation and energy-storage cost of the batteries in their performance analysis. In [16], a dynamic pricing algorithm is proposed in order to detect spiteful users and insecure energy providers. However, according to this solution, the users only consume energy and do not sell any energy from their DERs to an energy market. In [17], both consumers’ demand and energy providers’ profit are taken into account in a Stackelberg game in order to determine a daily-ahead hourly pricing. Similar to [14], the authors of [16] do not consider an energy broker, which plays a pivotal role in maximizing the profits of energy providers and consumers. In [18], the authors propose a fair dynamic pricing scheme to reduce peak demand based on real datasets and guarantee energy providers’ revenue. However, in their energy trading solution, consumers are not equipped with DERs. In [19], a dynamic pricing broadcasting scheme is proposed based on domestic load management. Authors consider that the energy providers have DER-based microgrids (MGs) and, similar to [13], the consumers do not participate in energy trading in their scheme. In [20], the authors introduce a day-ahead market where the market participants (consumers and energy providers) commit to buy or sell wholesale electricity one day prior to the operating day so as to avoid price volatility. Thus, in a day-ahead market, prices have a greater impact on demand. However, this study falls short in explaining how the demand of the consumers varies under different conditions (e.g., presence and the type of DERs). In [21], a usage-based dynamic pricing with privacy preservation for smart grid is proposed. The authors claim that as the proposed work enables the dynamic prices to take into account electricity usage, it can contribute to minimizing consumption costs while maximizing utility for both retailers and consumers. In [22], the authors propose a rolling horizon distributed probability control approach for demand-side management of smart grids using wind turbines. Unlike the solution in [22], where a distributed demand-side management was proposed, taking into consideration the wind power forecasting uncertainty, our solution introduced in this paper considers uncertainty in actual PV generation data, which is widely used in households. In [23], the authors present a multi-purpose probability optimization model for multiple MGs. In particular, optimal stochastic energy management is proposed to address multiple MGs structures with the aim of minimizing total cost and emissions, but our proposed solution embodies energy market-based modeling for energy trading.

In particular, the existing studies had some limitations, as the focus was on the benefits of energy trading only from the consumer’s perspective if the residual energy after the consumption was traded. Therefore, based on this premise, if the consumer trades the remaining energy into the REC where an energy provider can buy to reduce greenhouse gases (GHGs), the consumer essentially benefits from the energy trading. In addition, since energy providers can benefit from the reduction of carbon emissions by purchasing these RECs, REC trading benefits energy providers as well as consumers. Consequently, if existing studies were focused only on profit optimization from an energy seller’s perspective, the proposed study addressed demand and pricing optimization for both sellers (i.e., consumers) and buyers (i.e., energy providers) in consideration of REC, which has few previous studies.

In this paper, we investigate a dynamic pricing and demand control mechanism to maximize the profits of a single energy provider and consumers. The energy provider produces energy from its conventional energy sources (e.g., nuclear and thermal) and buys RECs from the DERs. The DER owners (i.e., consumers) also use ESSs in order to store energy for future use.

In particular, the contributions of this work are stated as follows:Contrary to the existing energy trading, we propose a new energy trading model facilitated by an energy broker, which also uses ESS in order to compensate for the fluctuating output from the energy generators. Particularly, the clean energy sources (DERs) and the conventional energy can be traded through the proposed energy broker where an energy provider and consumers participate in.To design the proposed energy trading model, we define an energy consumer model and energy provider model to maximize the benefits of energy consumers and providers. The energy consumer model includes a consumer utility function, an ESS cost function, and an RES cost function. On the other hand, the energy provider model considers a generation cost function and an RES trading profit function.In detail, we define demand and pricing decision problems considering RECs, RESs, ESSs, and their operation costs. First, to maximize the profit of the energy provider, we design an energy provider model, which decides optimal dynamic price for selling and the optimal amount of RECs that the energy provider ought to buy. Next, we design an energy consumer model to obtain the optimal energy demand from the energy provider while reducing the operation cost of RESs and ESSs.Via well-designed utility and functions, we formulate a demand-based optimization problem and optimal dynamic pricing decision to maximize the utilities of the energy provider and consumers. Using dual decomposition based on the convex optimization, we obtain the optimal Dynamic Selling through Dynamic Buying (DSDB) scheme, which runs at the energy broker in a day-ahead market.Finally, the results obtained through numerical simulation show that the proposed DSDB scheme outperforms the conventional pricing schemes. Furthermore, we evaluate our proposed DSDB using real-world energy market data where there exist various patterns of energy demand and energy generation.

The rest of this article is organized as follows. Section 2 presents details of our proposed demand-based trading mechanism, which includes the proposed energy provider model, consumer model, and optimization formulation. Numerical results with performance comparison depending on pricing strategy are presented in Section 3. Finally, we conclude the paper in Section 4.

## 2. Proposed Demand-Based Trading Mechanism with Dynamic Pricing Model of Renewable Energy Sources

We propose a demand-based trading mechanism with dynamic pricing for both selling and buying of energy in this section. We refer to our proposed scheme as DSDB in the subsequent part of this paper. According to the DSDB scheme, the selling and buying price of energy for a consumer is set dynamically by the energy broker in order to maximize total utility. In other words, with this proposed approach, consumers are charged a dynamic selling price and provided with a dynamic buying price by the energy broker. Using this scheme, the consumers can control their energy demand depending on the buying price as well as sell their own RES energy in the case when a higher selling price is offered. In this paper, we define RES energy as capacity (i.e., kW) per hour for RES.

Considering an energy broker market where consumers are capable of producing energy from their DERs and participate in energy trading, we propose our system model, as depicted in Figure 1. Consumers have DERs with PV-based energy generation capability and ESS for storing energy. When they are willing to sell their residual energy to the energy broker, they may provide REC voluntarily or they might be asked to provide it by the energy broker so as to comply with imposed environmental policies, similar to [11]. In our proposal, to supply energy, the energy provider may rely on consumers’ DERs besides its own conventional energy sources (e.g., nuclear and thermal). In other words, in our solution, the energy provider has only its conventional energy sources, and it may buy RECs when the energy from its own sources is not sufficient to meet the energy supply of the consumers at a given time. In our proposal, similar to [6], we consider that the consumers have a smart energy agent (SEA), which is a type of smart meter with the capability of communicating with other SEAs when it comes to trade energy. Additionally, we assume that an SEA allows not only a consumer to make the energy request but can also autonomously make an energy request to the energy broker if it is authorized by the consumer. In order to facilitate energy trading-related communication among the consumers and energy broker, we consider two messages: (i) energy request message, which incorporates the amount of required energy, and (ii) energy availability message, in which an energy provider states the amount of available energy it has and the per unit cost for trading.

The SEAs can be mounted on energy providers, energy brokers, and energy consumer sides. Therefore, information, such as power generation, energy price, consumption, and renewable energy trading can be defined as distributed local information in their respective areas, and global information can be updated over the communication network to update their local information again. Therefore, for energy brokers, the centralized framework processes the local information received from each SEA and provides updated information to each of the SEAs. As suggested in [7], the functionality of SEA can also change the information decision policy from a centralized to distributed manner. Therefore, the framework allows SEAs located anywhere in the system to participate in the management of energy trading technically and economically. The proposed solution can be deployed in both centralized and multi-agent framework-based SEA. It is worth note that both frameworks have their strengths and weaknesses in terms of privacy, computational and communication overhead, and implementation complexity. As suggested in [24], it is necessary to consider the tradeoff of both cases. One possible solution for determining the most appropriate one would be is to consider the number of participating agents. That is, a centralized framework can be applied when the number of components for energy trading is small, whereas a distributed case (multi-agent framework-based SEA) can be selected when the number of components is relatively larger.

The notations that are used for mathematical expressions presented in the subsequent part of this paper are tabulated in Table 1.

### 2.1. Energy Consumer Model

We formulate the quantity of energy used by the i-th consumer based on the k-th unit time slot as $x_{i,k}$. The lower and upper bound of energy consumption of the i-th consumer are described as $m_{i}$ and $M_{i}$, respectively. Therefore, we can describe the characteristic of $x_{i,k}$ as $m_{i}$ ≤ $x_{i,k}$ ≤ $M_{i}$. The energy from RESs is stored in the consumer premises, and the consumers have the authority to decide upon the amount of energy to resell to the energy broker for each unit time slot.

Then, let us define $g_{i,k}$ as the quantity of RES energy resold to the energy broker by the i-th consumer in the k-th unit time slot, and $L_{i}$ = $\left\{l_{i,t}|t=1,2,...,T\right\}$ clearly states the RES energy produced for the purpose of i-th consumer. We suppose that the produced curve function of RES generators is allowed. Since the quantity of resold RES energy cannot exceed the quantity of preserved energy, we can consider the inequality constraint condition of $g_{i,k}$ as follows:(1)$$0\leq g_{i,k}\leq \sum_{t=0}^{k-1}l_{i,t}-g_{i,t},\;\;\forall i\in N,k\in K.$$

The RES energy is resold here, which implies that the PVs of consumers have produced energy during the day. The consumer then produces and spends for its own energy and stores the remaining energy in the ESS. It also sells the remaining ESS energy to the energy broker during high-priced hours. Furthermore, we can take into account the stored energy capacity of an ESS as $C_{i}$. We can also consider the quantity of energy preserved in the k-th unit time slot as $c_{i,k}$. Finally, we can describe the quantity of energy filled up from the electric grid ($e_{i,k}$≥ 0) or discharged to the electric grid ($e_{i,k}$≤ 0) by the i-th consumer in the k-th unit time slot as $e_{i,k}$. Therefore, we can formulate the quantity of energy in the energy storage as
(2)$$c_{i,k}=\sum_{t=0}^{k}e_{i,k},\;\forall i\in N,k\in K.$$

The ESS of the i-th consumer, in general, has a maximum charging and discharging rate defined as $e_{i,max}$ and $-e_{i,min}$, respectively. Therefore, we have the inequality constraints as follows:(3)$$0\leq c_{i,k}\leq C_{i},\;\forall i\in N,k\in K,$$
(4)$$-e_{i,min}\leq e_{i,k}\leq e_{i,max},\;\forall i\in N,k\in K.$$

#### 2.1.1. Consumer Utility Function

We can formulate the different consumer behavior based on a differentiated selection of various utility-based functions [25]. We can describe U(x,δ,s) as the equivalent utility function for all consumers. Here, *x* is the level of energy usage for the consumer, δ is the variable of maximum charging of an ESS, which may differ among consumers at dissimilar unit times of a given day, and *s* is the quantity of the energy selling level of the consumer. Specifically, the utility function describes the level of consumer satisfaction that consumers experience from charging ESS and selling energy as a function of its energy usage for each consumer. Therefore, we suppose that the consumer utility functions meet the following properties according to different consumers.

(I)Utility function is nondecreasing. This implies mathematically that
(5)$$\frac{\partial U(x,\delta ,s)}{\partial x}\geq 0.$$(II)A marginal benefit function is defined as
(6)$$V\left(x,\delta ,s\right)\dot{=}\frac{\partial U(x,\delta ,s)}{\partial x}.$$The above marginal benefit of a consumer is nonincreasing. Therefore, it can be rewritten as
(7)$$\frac{\partial V(x,\delta ,s)}{\partial x}\leq 0.$$That is to say, the utility function is concave. This implies that the satisfaction level of the consumers can gradually become saturated.(III)We assume that without energy consumption, there is no utility. Namely, we can present this as
(8)U(0,δ,s)=0,∀δ,s>0.

There are various choices of utility functions in the literature to formulate the satisfaction of energy consumers. From the aspect of microeconomics, we model the reactions of different consumers by using the concept of utility function [25]. Here, we define $U(x_{i,k},\delta_{i,k},s_{i,k})$ as the utility function for each consumer, where $x_{i,k}$ indicates the quantity of energy used within the k-th unit time slot in the i-th consumer, $\delta_{i,k}$ represents the quantity of maximum charging of the ESS in the k-th unit time slot by the i-th consumer, and $s_{i,k}$ represents the quantity of energy selling to energy broker in the k-th unit time slot by the i-th consumer. We can take the different utility functions by choosing a disparate value of $\delta_{i,k}$. To be specific, the utility function $U(x_{i,k},\delta_{i,k},s_{i,k})$ represents the level of satisfaction for each consumer with the quantity of charging of ESS $\delta_{i,k}$, energy $x_{i,k}$ consumed, and energy sold to the energy broker in the k-th time slot by the i-th consumer. We assume that the ESS is used for reducing the demand request to the energy broker and that the quantity of energy consumed by the consumer does not exceed the capacity of the ESS. Therefore, we assume that the i-th consumer will have the highest satisfaction when its consumption reaches the ESS capacity. Furthermore, even if it consumes more than that, the satisfaction remains constant as there is no energy stored in the ESS. Here, we apply a utility function with a quadratic function [26,27], defined as
(9)$$\begin{array}{cc} \; & U\left(x_{i,k},\delta_{i,k},s_{i,k}\right)=\left\{\begin{array}{cc} w_{i}x_{i,k}-\frac{w_{i}}{2\delta_{i,k}}x_{i,k}^{2}-s_{i,k},\;\mathrm{if}\;0\leq x_{i,k}\leq \delta_{i,k}, & \\ \frac{w_{i}\delta_{i,k}}{2}-s_{i,k},\;\;\;\mathrm{if}\;x_{i,k}\geq \delta_{i,k}, \end{array}\right. \end{array}$$
where $x_{i,k}$ indicates the quantity of energy consumed by k-th unit time slot in the i-th consumer, $w_{i}$ (i.e., demand weight factor) is the i-th given consumer’s preference, $\delta_{i,k}$ is the quantity of maximum charging of ESS in the k-th unit time slot by the i-th consumer, and $s_{i,k}$ is the quantity of energy sold to the energy broker in the k-th unit time slot by the i-th consumer.

#### 2.1.2. Cost Function of ESS Operation

The ESS operation cost depends on how much and how speedily it charges and discharges energy. Consequently, it is assumed that the ESS operational cost function $C_{ESS}$ presents a convex function of $e_{i,k}$ [28]. The ESS operational cost function with the parameters ξ(ξ>0) and κ(κ≥0) is
(10)$$C_{ESS}\left(e_{i,k}\right)=\xi e_{i,k}^{2}+\kappa ,$$
(11)$$d_{i,k}=\sum_{l=1}^{k}e_{i,l},\;0\leq d_{i,k}\leq D_{i},$$
where $e_{i,k}$ is the quantity of energy charging and discharging by the k-th unit time slot by the i-th consumer, ξ is the slope coefficient parameter (e.g., slope of a battery type) and κ is the initial energy level of the battery, $d_{i,k}$ is the quantity of energy stored in the k-th unit time slot by the i-th consumer, and $D_{i}$ is ESS capacity of the i-th consumer.

**Lemma** **1.**
*The proposed $C_{ESS}\left(e_{i,k}\right)$ has a convex function and a unique solution for the quadratic ESS operational cost model.*


**Proof.** From Equation (Equation 10), we can rewrite
$$C_{ESS}\left(e_{i,k}\right)=\xi e_{i,k}^{2}+\kappa ,$$
where $e_{i,k}>0$, ξ>0 and κ≥0. For $C_{ESS}\left(e_{i,k}\right)$, the ESS operational cost function stems from $\frac{\partial }{\partial e_{i,k}}C_{ESS}\left(e_{i,k}\right)>0$, $\frac{\partial^{2}}{\partial^{2}e_{i,k}}C_{ESS}\left(e_{i,k}\right)>0$. Therefore, the proposed ESS operational cost function $C_{ESS}\left(e_{i,k}\right)$ is a convex function and a unique solution for $e_{i,k}$. □

#### 2.1.3. Cost Function of RES Energy

The selling cost of the energy from a RES generator is described as a function $C_{RES}\left(g_{i,k}\right)$ associated with the quantity of energy sold. In particular, this cost is determined based on two factors as we can notice from (Equation 12). The first and second terms in (Equation 12) are the operation cost of an ESS and the maintenance and repayment installation cost of the RES generator during its continuous cycle of life, respectively.

Therefore, the cost function of the RES energy is formulated as follows:(12)$$C_{RES}\left(g_{i,k}\right)=\zeta g_{i,k}^{2}+\eta g_{i,k},$$
(13)$$0\leq g_{i,k}\leq \sum_{l=1}^{k-1}p_{i,l}-g_{i,l},$$
where $g_{i,k}$ is the quantity of the RES energy sold to the energy broker by the k-th unit time slot by the i-th consumer, ζ is a slope coefficient parameter (e.g., slope of battery type) (ζ>0), η is average cost per unit of the RES energy and $p_{i,l}$ is the RES energy output for the i-th consumer.

### 2.2. Energy Provider Model

We indicate the energy generated from the energy provider within the k-th unit time slot as $G_{k}$. The energy provider furnishes a lower bounded energy $G_{k,min}$ to fulfill the overall smallest energy demands of all consumers in the k-th unit time slot. In addition, the energy provider also guarantees an upper bounded energy $G_{k,max}$ to include the largest demands of all consumers to avoid system blackout due to energy deficiency. Moreover, the energy provider determines how much energy it needs to generate and how much RES energy it needs to repurchase. We describe $R_{k}$ as the quantity of RES energy that the energy provider purchases from consumers in the k-th unit time slot.

#### 2.2.1. RES Energy Trading Profit Function

We present how the energy provider can turn carbon footprint reductions into substantial profits through RECs trading. The Kyoto Protocol grant offsetting is a way for government and private enterprises to obtain carbon credits that they could trade in the marketplace [29]. Carbon emissions can also be addressed through RES systems, the shape of RECs or the RES energy purchased by utilities, and carbon offsets by GHG reporting programs. Therefore, a profit function for RECs trading can be summarized as
(14)$$F\left(M_{k}\right)=P\times M_{k},$$
where $F(M_{k})$ denotes the profit function in the k-th unit time slot, *P* indicates the price value per unit carbon emissions (*P* > 0), and $M_{k}$ shows the quantity of carbon footprint in the k-th unit time slot. The relationship of *P* and $M_{k}$ based on financial market-based trading theory [30] can be expressed as
(15)$$P=-\lambda M_{k}+\mu .$$

Equation (Equation 15) indicates the market price of energy falls when supply increases. This can be again stated as follows:(16)$$F\left(M_{k}\right)=-\lambda \tau^{2}R_{k}^{2}+\mu \tau R_{k},$$
where $M_{k}$ is the quantity of carbon emission $\left(M_{k}>0\right)$, $R_{k}$ is $R_{k}$ unit RES energy $\left(R_{k}>0\right)$ in the k-th unit time slot, λ, μ, and τ are slope coefficient parameters (λ,μ,τ>0), respectively.

The relationship between carbon emissions and RES energy is represented as $M_{k}=\tau R_{k}$. This shows we are able to lessen $\tau R_{k}$ unit carbon emission through purchasing $R_{k}$ unit REC. For example, energy utilities in the Republic of Korea emit 0.517 kg of carbon in producing 1 kWh of energy [31]. The energy provider lessens their comprehensive carbon emissions by 0.517 kg by purchasing 1 kWh of energy from RESs. Therefore, the profit function relevant to the quantity of RES energy can be presented as
(17)$$F\left(R_{k}\right)=-\lambda R_{k}^{2}+\mu R_{k}.$$

#### 2.2.2. Generation Cost Function

We describe $C_{G}\left(G_{k}\right)$ as the cost for the energy provider to generate $G_{k}$ energy. The cost function is modeled as [27,28,32]
(18)$$C_{G}\left(G_{k}\right)=\alpha G_{k}^{2}+\beta G_{k}+\gamma ,$$
where $G_{k}$ is the energy generated by the energy provider in the k-th time slot. α, β, and γ is quadratic/linear/no load coefficient parameter (α>0,β≥0,γ≥0), respectively.

### 2.3. Demand-Based Optimization Problem Formulation

In this subsection, we formulate the demand-based optimization problem [33]. To find a solution to the problem efficiently in a convex dual manner, the primal problem is taken apart into a consumer subproblem and an energy provider subproblem. After that, we propose two dual mechanisms for each part. In the viewpoint of consumers’ demand, we can express a unit time-slot-dependent demand-based trading optimized formulation, which is a convex problem that maximizes the total utility. That is to say, the total utility means that the total profit of all consumers increases as the profit of the energy provider increases. Therefore, we can state the following optimization problem under several constraints:(19)$$\begin{array}{cc} {\mathrm{maximize}}_{H}\; & \sum_{i}^{N}\sum_{k}^{K}(\omega U\left(x_{i,k},\delta_{i,k},s_{i,k}\right)-\psi C_{ESS}\left(e_{i,k}\right)-C_{RES}\left(g_{i,k}\right))\\ \; & +\sum_{k}^{K}(F\left(R_{k}\right)-\chi C_{G}\left(G_{k}\right)), \end{array}$$
(20)$$\begin{array}{cc} \mathrm{subject}\;\mathrm{to}\; & \sum_{i}^{N}\left(x_{i,k}+e_{i,k}-g_{i,k}\right)\leq G_{k},\;\forall k\in K, \end{array}$$
(21)$$\;\;\sum_{i}^{N}g_{i,k}=R_{k},\;\forall k\in K,$$
where ***H*** = $\left\{x_{i,k},\delta_{i,k},s_{i,k},e_{i,k},g_{i,k},R_{k},G_{k}|i\in N,k\in K\right\}$, ω is a weight factor for profit per energy consumption, ψ is a weight factor for the ESS price ratio, and χ is a weight factor for the ESS capacity ratio.

The inequality constraint in (Equation 20) shows that the net demand of all consumers cannot go over the energy supply limit from the energy provider in each unit time slot. The equality constraint in (Equation 21) presents that the total selling quantity of RES energy for all consumers should be equal to the purchasing quantity of the energy providers in every unit time slot.

In addition, firstly, all elements of the objective function in (Equation 19) represent concave properties or convex properties. Secondly, inequality and equality constraints in (Equation 20) and (Equation 21) denote linear functions. Thirdly, the feasible set represents a convex set. Finally, the objective function in (Equation 19) shows the concave function. Thus, the proposed problem can be resolved centrally by convex programming manners as a convex optimization problem. Furthermore, to ensure consumer privacy, we will solve the demand-based trading problem in a convex dual decomposition scheme [15,33,34,35].

#### Dual Decomposition to Primal and Subproblem

To effectively solve the problem in a dual decomposition scheme, the primal problem is transformed to a dual problem through a Lagrange function. Finally, the Lagrange dual function for dual decomposition is proposed. Therefore, we require to modify the inequality and equality constraints in (Equation 20) and (Equation 21), respectively, so as to differentiate between the consumer and the energy provider by using Lagrange multipliers. Thus, by applying this technique, we can easily solve the problem in (Equation 19).

Here, firstly, we require to reconfigure the parameters $x_{i,k}$, $e_{i,k}$, $g_{i,k}$, and $G_{k}$ in inequality constraint (Equation 20) and parameters $g_{i,k}$ and $R_{k}$ in equality constraint (Equation 21). For the decomposition of the primal problem of (Equation 19), we can describe the Lagrangian as [15,33,34,35],
(22)$$\begin{array}{c} \begin{array}{cc} L(H,\nu ,o)=\; & \sum_{i}^{N}\sum_{k}^{K}(\omega U\left(x_{i,k},\delta_{i,k},s_{i,k}\right)-\psi C_{ESS}\left(e_{i,k}\right)-C_{RES}\left(g_{i,k}\right))\\ \; & +\sum_{k}^{K}(F\left(R_{k}\right)-\chi C_{G}\left(G_{k}\right))-\sum_{k}^{K}\nu_{k}(\sum_{i}^{N}\left(x_{i,k}+e_{i,k}-g_{i,k}\right)-G_{k})\\ \; & +\sum_{k}^{K}o_{k}(\sum_{i}^{N}g_{i,k}-R_{k}), \end{array} \end{array}$$
where ν and o are Lagrange multipliers, which are defined as $\left\{\nu_{k}|k\in K\right\}$ and {$o_{k}$∣k∈K}, respectively.

Moreover, the Lagrange dual function is expressed in (Equation 23).
(23)$$g\left(\nu ,o\right)={\mathrm{maximize}}_{H}L\left(H,\nu ,o\right),$$
where the constraints (Equation 20) and (Equation 21) are mitigated by Lagrangian (Equation 22), and we can revise (Equation 22) as
(24)$$\begin{array}{cc} L(H,\nu ,o)=\; & \sum_{i}^{N}\sum_{k}^{K}(\omega U\left(x_{i,k},\delta_{i,k},s_{i,k}\right)-\psi C_{ESS}\left(e_{i,k}\right)-C_{RES}\left(g_{i,k}\right)-\nu_{k}\left(x_{i,k}+e_{i,k}-g_{i,k}\right)+o_{k}g_{i,k})\\ \; & +\sum_{k}^{K}(F\left(R_{k}\right)-\chi C_{G}\left(G_{k}\right)+\nu_{k}G_{k}-o_{k}R_{k}). \end{array}$$

Due to the individuality of consumers, the duality function in (Equation 23) can be reformulated as
(25)$$\begin{array}{cc} g(\nu ,o)=\; & \sum_{i}^{N}\max\sum_{k}^{K}(\omega U\left(x_{i,k},\delta_{i,k},s_{i,k}\right)-\psi C_{ESS}\left(e_{i,k}\right)-C_{RES}\left(g_{i,k}\right)-\nu_{k}\left(x_{i,k}+e_{i,k}-g_{i,k}\right)+o_{k}g_{i,k})\\ \; & +\max\sum_{k}^{K}(F\left(R_{k}\right)-\chi C_{G}\left(G_{k}\right)+\nu_{k}G_{k}-o_{k}R_{k}). \end{array}$$

For clarity, (Equation 25) can be organized as
(26)$$\begin{array}{cc} g(\nu ,o)=\; & \sum_{i}^{N}\{\max\sum_{k}^{K}(\omega U\left(x_{i,k},\delta_{i,k},s_{i,k}\right)-\psi C_{ESS}\left(e_{i,k}\right)-C_{RES}\left(g_{i,k}\right)-\nu_{k}\left(x_{i,k}+e_{i,k}-g_{i,k}\right)+o_{k}g_{i,k})\}\\ \; & +\max\sum_{k}^{K}(F\left(R_{k}\right)-\chi C_{G}\left(G_{k}\right)+\nu_{k}G_{k}-o_{k}R_{k}). \end{array}$$

For simplicity, (Equation 26) can be rewritten as
(27)$$\begin{array}{cc} g(\nu ,o)=\; & \sum_{i}^{N}\Pi_{i}\left(\nu ,o\right)+\Theta \left(\nu ,o\right), \end{array}$$
where
(28)$$\begin{array}{cc} \Pi_{i}\left(\nu ,o\right)=\; & \max\sum_{k}^{K}(\omega U\left(x_{i,k},\delta_{i,k},s_{i,k}\right)-\psi C_{ESS}\left(e_{i,k}\right)-C_{RES}\left(g_{i,k}\right)-\nu_{k}\left(x_{i,k}+e_{i,k}\right)+\left(\nu_{k}+o_{k}\right)g_{i,k}), \end{array}$$
(29)$$\begin{array}{cc} \Theta (\nu ,o)=\; & \max\sum_{k}^{K}(F\left(R_{k}\right)-\chi C_{G}\left(G_{k}\right)+\nu_{k}G_{k}-o_{k}R_{k}). \end{array}$$

At this point, a consumer subproblem (Equation 28) and an energy provider subproblem (Equation 29) have been taken apart from the Lagrange dual function. Moreover, by setting buying price $\nu_{k}$ and selling price ($\nu_{k}$ + $o_{k}$) in the aspect of the consumer’s view, subproblems (Equation 28) and (Equation 29) are the same as the consumer welfare function and the energy provider profit function, respectively. Consequently, we can write the Lagrange dual problem as follows:(30)$$D\left(\nu ,o\right)=\min_{\nu \geq 0,o}g\left(\nu ,o\right).$$

Here, we convert the primal problem (Equation 19) to the dual problem (Equation 30). After that, we pursue a solving method with respect to the dual problem. Since the primal problem indicates convex function and the slater’s condition is retained for the primal problem, strong duality holds. Thus, the duality gap indicates 0.

Consequently, instead of finding a solution to a primal problem, we can find a solution to a dual problem.

### 2.4. Gradient Iteration Method for Demand-Based Trading

In this subsection, we design our demand-based trading mechanism with a dynamic pricing model of RESs to find a solution concerning the dual problem. Here, we can find a solution to the problem iteratively using the gradient projection method [36] as follows:(31)$$\begin{array}{cc} \nu^{l+1}\; & ={\left[\nu^{l}-\tau \frac{\partial g(\nu ,o)}{\partial \nu }\right]}^{+}\\ \; & ={\left[\nu^{l}-\tau (\sum_{i}^{N}\left(g_{i}^{*}\left(\nu^{l},o^{l}\right)-x_{i}^{*}\left(\nu^{l},o^{l}\right)-e_{i}^{*}\left(\nu^{l},o^{l}\right)\right)+G^{*}\left(\nu^{l},o^{l}\right))\right]}^{+}, \end{array}$$
(32)$$\begin{array}{cc} o^{l+1}\; & =\left[o^{l}-\tau \frac{\partial g(\nu ,o)}{\partial o}\right]\\ \; & =\left[o^{l}-\tau (\sum_{i}^{N}\left(g_{i}^{*}\left(\nu^{l},o^{l}\right)\right)-R^{*}\left(\nu^{l},o^{l}\right))\right]. \end{array}$$

Here, l∈ρ, ρ presents the set of repetition parameters, and τ indicates the step size in the gradient method. We indicate $x_{i}^{*}\left(\nu^{l},o^{l}\right)$, $g_{i}^{*}\left(\nu^{l},o^{l}\right)$ and $e_{i}^{*}\left(\nu^{l},o^{l}\right)$ as the optimization tool of subproblem (Equation 28) concerning i-th consumer, and $R^{*}\left(\nu^{l},o^{l}\right)$, $G^{*}\left(\nu^{l},o^{l}\right)$ as the optimization tool of subproblem (Equation 29) with a given $\nu^{l}$, $o^{l}$ concerning energy provider.

Thus, we can present two kinds of algorithms to seek a solving method of the dual problem that represents the consumer and energy provider aspects. Initially, Algorithm 1 decides to acquire the values of $x_{i}^{*}\left(\nu^{l},o^{l}\right)$, $g_{i}^{*}\left(\nu^{l},o^{l}\right)$ and $e_{i}^{*}\left(\nu^{l},o^{l}\right)$ by solving the subproblem in (Equation 28) given pricing information $\nu^{l}$ and $o^{l}$ from the energy provider. Second, Algorithm 2 decides the values of $R^{*}\left(\nu^{l},o^{l}\right)$ and $G^{*}\left(\nu^{l},o^{l}\right)$ by solving subproblem in (Equation 29) through the response information $x_{i}^{*}\left(\nu^{l},o^{l}\right)$, $g_{i}^{*}\left(\nu^{l},o^{l}\right)$ and $e_{i}^{*}\left(\nu^{l},o^{l}\right)$ obtained from every consumer in addition to the updated values of $\nu^{l}$ and $o^{l}$.
**Algorithm 1** Consumer’s Demand Decision Algorithm1:**procedure**Consumer Demand  ($\nu^{l},o^{l}$)2: **while** each l∈ρ **do**3:  get $x^{*}\left(\nu^{l},o^{l}\right)$, $g^{*}\left(\nu^{l},o^{l}\right)$, $e^{*}\left(\nu^{l},o^{l}\right)$ by solving subproblem (Equation 28) for the given $\nu^{l},o^{l}$4:  send consumer’s demand $x^{*}\left(\nu^{l},o^{l}\right)$, $g^{*}\left(\nu^{l},o^{l}\right)$, $e^{*}\left(\nu^{l},o^{l}\right)$ to the energy provider5: **end while**6: **return** $x^{*}\left(\nu^{l},o^{l}\right)$, $g^{*}\left(\nu^{l},o^{l}\right)$, $e^{*}\left(\nu^{l},o^{l}\right)$7:**end procedure**

**Algorithm 2** Energy Provider’s Dynamic Pricing Decision Algorithm
1:**procedure**Pricing Information  ($x^{*}$, $g^{*}$, $e^{*}$)2: **while** each l∈ρ **do**3:  adjust $R^{*}\left(\nu^{l},o^{l}\right)$, $G^{*}\left(\nu^{l},o^{l}\right)$ by solving subproblem (Equation 29)4:  update pricing data as follows5:  $\nu^{l+1}={\left[\nu^{l}-\tau \left(\sum_{i}^{N}\left(g_{i}^{*}\left(\nu^{l},o^{l}\right)-x_{i}^{*}\left(\nu^{l},o^{l}\right)-e_{i}^{*}\left(\nu^{l},o^{l}\right)\right)+G^{*}\left(\nu^{l},o^{l}\right)\right)\right]}^{+}$6:  $o^{l+1}=\left[o^{l}-\tau \left(\sum_{i}^{N}\left(g_{i}^{*}\left(\nu^{l},o^{l}\right)\right)-R^{*}\left(\nu^{l},o^{l}\right)\right)\right]$7:  dispatch updated pricing data to every consumer8: **end while**9: **return** $\nu^{l+1},o^{l+1}$10:
**end procedure**



### 2.5. Overall Demand-Based Trading Mechanism

In short, the overall demand-based trading mechanism with RESs works as follows: From the first hour of the day, the energy provider firstly establishes a selling price $\nu^{0}$ and a repurchasing price ($\nu^{0}+o^{0}$), which are announced to the consumers through the energy broker. After receiving the information, every consumer makes an attempt to maximize their own profit with relevant price data and sends a corresponding response to the energy provider. After reflecting the response information of consumers’, the energy provider adjusts new prices to maximize profits, and then dispatches the updated new dynamic pricing data to all the consumers. Therefore, the dynamic pricing strategy procedure makes up an iterative linkage structure between consumers and the energy provider. If this linked structure is convergent at the starting point of a given day, the selling price ν and the repurchasing price (ν+o) are secured.

Hence, if the energy provider configures the selling price as ν and the repurchasing price as (ν+o), the consumers maximize their profits. Therefore, dynamic pricing and demand decision sequence flows in Algorithms 1 and 2 present interactive interactions between the energy provider and consumers, as depicted in Figure 2.

## 3. Performance Evaluation and Discussion

We perform intensive simulations to indicate that the proposed method is able to accomplish demand-based energy trading for the grid and total utility for the consumers.

### 3.1. Simulation Assumption

The proposed simulated demand-based trading system environment comprises an energy provider and 100 residential consumers, and each of them is equipped with a PV and an ESS. We suppose that all the consumers install and operate a 5 kW PV system and a 6 kW ESS at their residential areas. The average generating cost of a 5 kW PV system is around 79.93 Korea won (KRW) per kWh (USD 0.0687 per kWh), and the battery capacity of an ESS is 6 kWh for residential areas. Although the our mathematical formulation can be extended by considering the residential areas with different ESS capacities, for the sake of simplicity, we suppose that the ESSs deployed in residential areas have the same capacity in this performance evaluation. At this point, it is worth noting that in a real deployment scenario, the residential users deploy lithium-ion-based accumulator batteries in order to attain desired capacity. According to [37], the price of such batteries would be 1500 USD/kWh.

Figure 3 demonstrates the electricity output performance on a clear sunny day of a PV that is installed in an electronics and telecommunications research institute testbed [38,39], Republic of Korea. From this figure, we notice that the PV provides at least 1 kWh of output power of around 33% of a day.

We assume that every consumer has different minimal and maximal bound of energy demand requirement. The decision variables we consider here have real values. Each consumer’s preference *w* is arbitrarily decided within a range from 0.5 to 6.5. Additionally, in our performance evaluation, we use real market data from independent electricity system operator market report [40] in order to present the consumers’ energy demand pattern on a given day, as depicted in Figure 4 as the input of 100 residential consumers’ demand preference. Demand preference means that consumers with higher preference are likely to use more energy.

### 3.2. Demand-Side-Based Resource Operation Performance Comparison

We measure the total energy demand of consumers under three different cases, with different resource configurations on the consumer side: (i) the consumers do not have both PV and ESS, (ii) the consumers have only ESS, and (iii) the consumers have both PV and ESS.

As depicted in Figure 5, the third case relatively reduces more peak-time energy demand than other cases because consumers are able to resell RES energy to the energy broker when they are equipped with both ESS and PV. The results in Figure 6 indicate that the third case shows no sharp fluctuation in demand, resulting in the smallest variance in consumer demand. Therefore, in this case, the consumers’ demand is adequately satisfied, and they will have an uninterrupted power supply due to having their own PV and ESS. Figure 7 presents that the third case has the smallest peak-to-average ratio as well. Additionally, we notice from this figure that, in this case, the consumers’ demand during peak-times does not fluctuate abruptly. This indicates that a power outage is less likely to occur in this case. Therefore, we conclude that when consumers are equipped with both PV and ESS, they can have more stable energy supply compared to the other two cases.

### 3.3. Analysis of Various Demand-Based Trading Strategies

According to our proposed DSDB scheme, a consumer has a dynamic selling and buying price of energy, which is decided by the energy broker. In this subsection, we want to evaluate this proposed scheme with three other conventional schemes, which are stated below:(I)Fixed Selling through Fixed Buying (FSFB) scheme: According to this scheme, the consumers are charged with a fixed selling price and provided with a fixed buying price [9], which is the most common scheme in conventional smart grids. In this strategy, consumers are provided with energy based on personal preferences without taking the price into account. Therefore, there is no incentive for consumers to manage the timing of reselling RES energy to the energy broker deliberately.(II)Fixed Selling through Dynamic Buying (FSDB) scheme: This strategy charges consumers a fixed selling price and provides consumers with a dynamic buying price for RES energy.(III)Dynamic Selling through Fixed Buying (DSFB) scheme: This strategy allows the consumers a dynamic selling price and provides them with a fixed buying price for RES energy [28]. In this strategy, consumers do not store RES energy for later selling but prefer to sell the residual energy instantly at a dynamic price.

Next, we evaluate DSDB (proposed scheme), DSFB, FSDB and FSFB under the case where the consumers are equipped with both PV and ESS. Furthermore, we analyze those four strategies of pricing based on the proposed total utility.

In this performance evaluation, we consider that the energy broker is equipped with an ESS. When the energy provider produces more energy than the consumers’ demand, the residual energy will be discarded inefficiently. Currently, an ESS is expensive to install, but the cost of its installation will gradually decline with technological advancements in the foreseeable future. Bearing this in mind, we consider that the energy broker will be equipped with a low-cost ESS in the coming days. This will allow the energy broker to store the residual energy and sell it to the consumers when necessary. Considering that the energy brokers would have low-cost ESS for storing energy in the future, we want to evaluate those four strategies.

First of all, for evaluating those four strategies, we consider the current installation cost of an ESS. The installation cost of an ESS 1 MWh is about KRW 540 million in the Republic of Korea (USD 464,117) [41]. Figure 8 demonstrates the total utility performance for DSDB, FSFB, DSFB, and FSDB. The total utility is quantified by summing up the value of the overall profit during each unit time slot in 24 h. The summary of the results, which is stated in Table 2, indicates that DSDB outperforms the other three strategies (see the first row).

In view of this case when ESS cost would be reduced in the future, let us consider that the installation cost of an ESS 1 MWh is about KRW 5 million in the Republic of Korea (USD 4297). With the drop of an ESS price, we surmise that the number of energy brokers with ESS will increase significantly in the future. Considering such a case, we present the total utility of those four strategies in Figure 9. In this case, the total utility is measured taking into account the installation cost of an ESS. The summary of the result is presented in Table 2 (see the second row). We can notice from this table that the highest profit (229.1 cents) is achieved when DSDB is applied, while the lowest profit (200.1 cents) can be made when FSFB is used.

In a nutshell, in both cases (i.e., an energy broker with low-cost and high-cost ESS), our proposed DSDB provides the highest total utility among all the strategies. However, looking at this table, one can realize that when an ESS price is low, DSDB leads to ensuring 3.52% more profit compared to the case when the price is high.

## 4. Conclusions

In this paper, we have proposed an optimal demand and dynamic pricing mechanism in order to maximize the profit of consumers by increasing the profit of an energy provider in an energy trading ecosystem where clean energy generation is encouraged through RECs trading. We formulated convex optimization problems using dual decomposition in order to decide the optimal demand and dynamic pricing. Considering the present and future deployment cost of ESS in an energy broker, we have rigorously studied the proposed DSDB scheme and observed that it noticeably increases the profit of both the consumer and energy provider no matter how the price of ESSs changes. In the proposed mechanism, regardless of the cost of ESS, the total utility of DSDB is approximately 15% higher than that of the existing FSFB. As a result, we found that the proposed mechanism in terms of total utility is excellent when adjusting demand with dynamic prices. The limitation of this proposed study is that when the number of agents increases, it becomes more complicated to calculate as a centralized case, and there is also a disadvantage that the decision of the target to be traded will be delayed in real-time. In addition, distributed cases have the disadvantage of having to pay for the use of communication overheads to solve problems by exchanging local information with other SEAs, so structural studies on overcoming these limitations are also necessary.

In the future, the spread of DER is expected to lead to the basic installation and use of PV and ESS for each household, and the proposed mechanism will help consumers benefit from an economical reduction in energy supply costs. Therefore, our proposed mechanism is expected to be used as an IoT application to the large scale energy trading market where existing energy providers and consumers participate more actively.

## Figures and Tables

**Figure 1 sensors-21-05819-f001:**
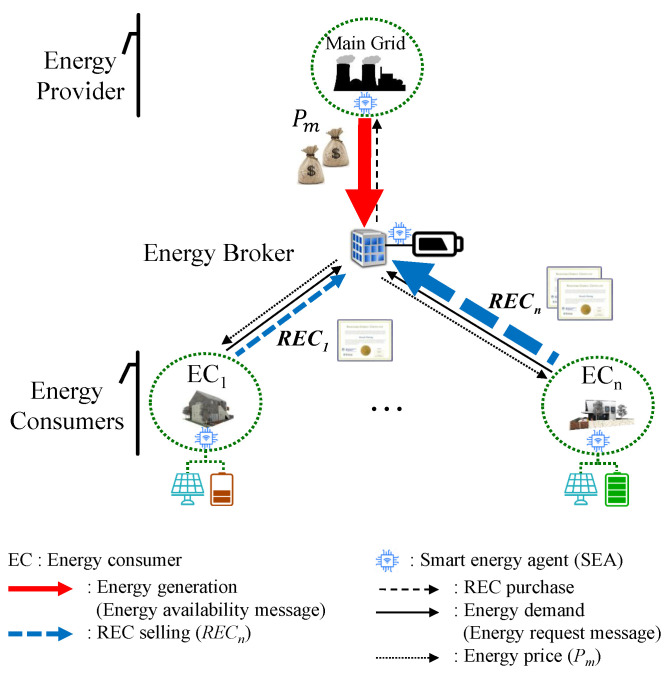
System model of proposed demand-based trading mechanism in an energy broker with dynamic pricing.

**Figure 2 sensors-21-05819-f002:**
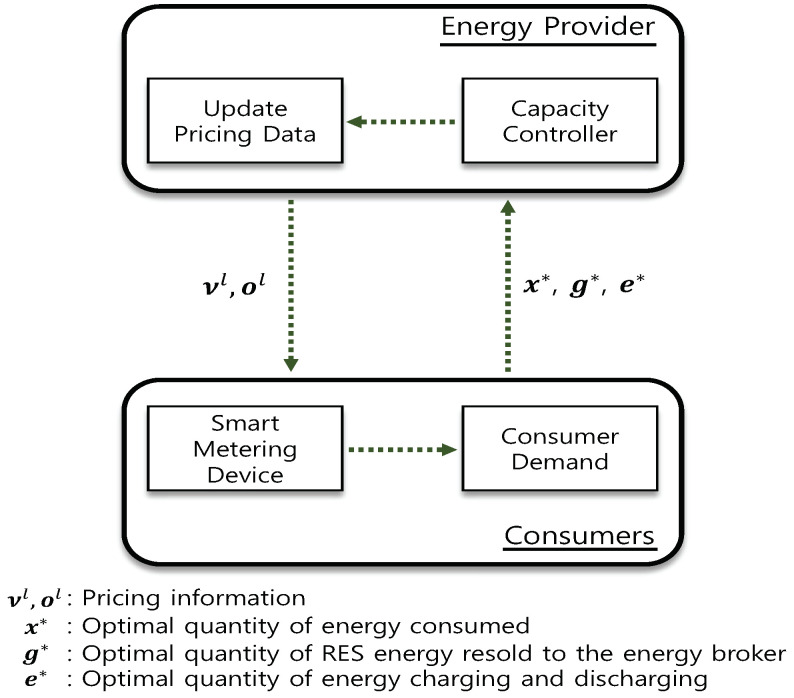
Interactions between the energy provider and consumers.

**Figure 3 sensors-21-05819-f003:**
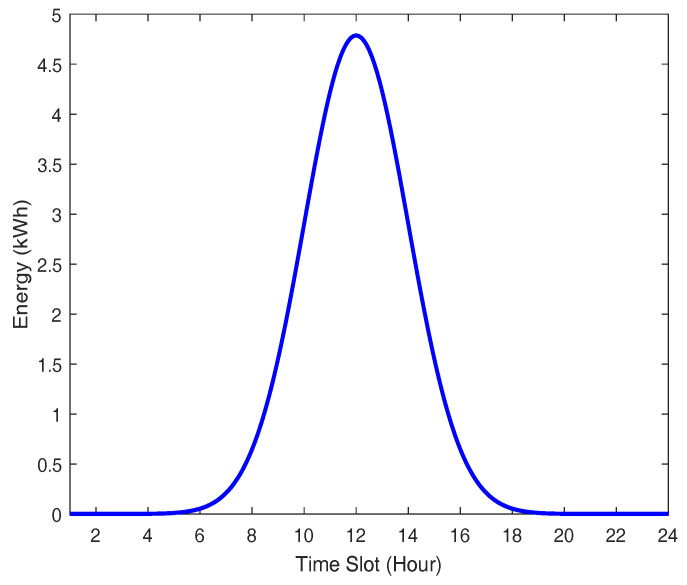
Electricity output curve according to PV power generation.

**Figure 4 sensors-21-05819-f004:**
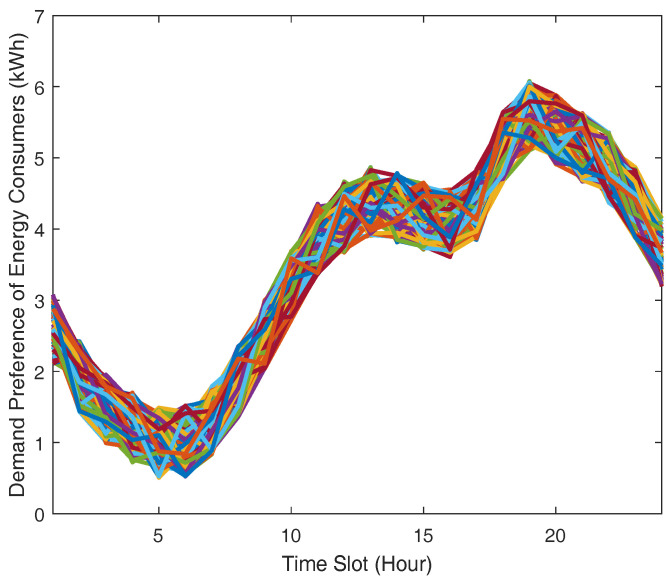
Demand preferences of consumers.

**Figure 5 sensors-21-05819-f005:**
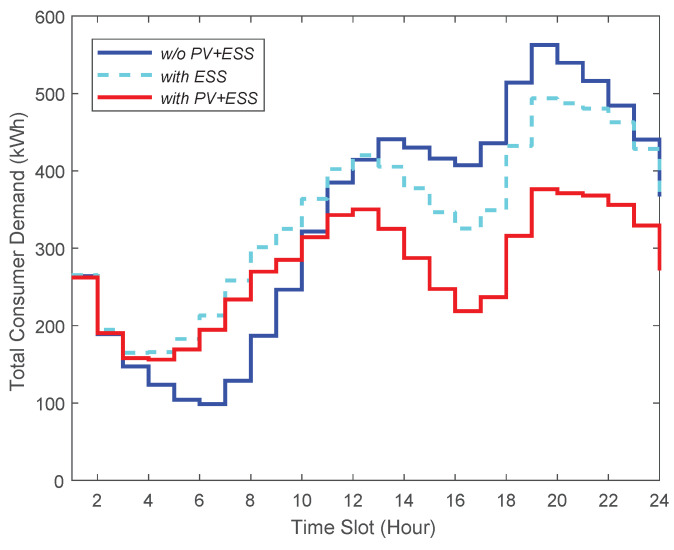
Comparison of total consumer demand.

**Figure 6 sensors-21-05819-f006:**
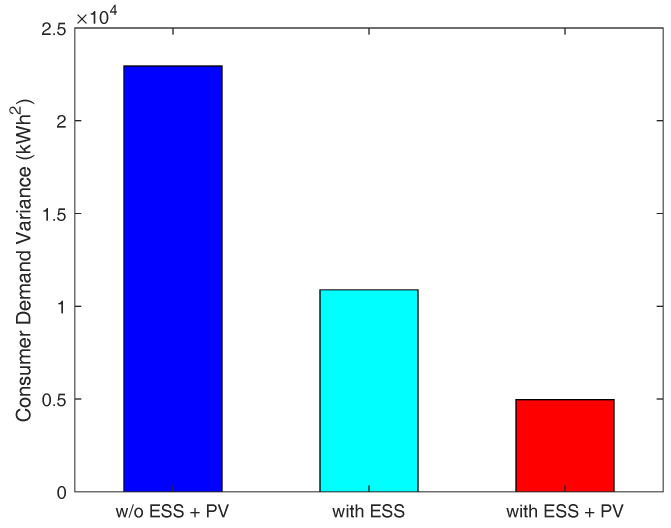
Consumer demand variance.

**Figure 7 sensors-21-05819-f007:**
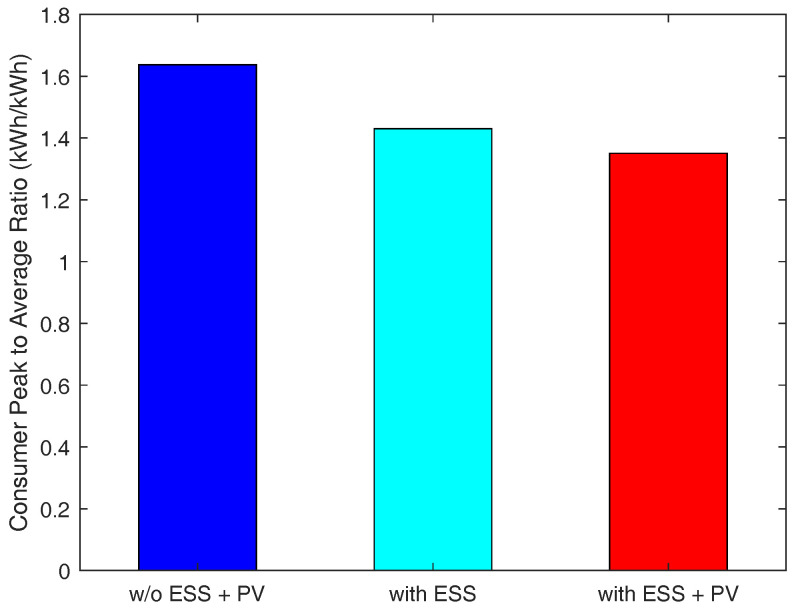
Consumer peak to average ratio.

**Figure 8 sensors-21-05819-f008:**
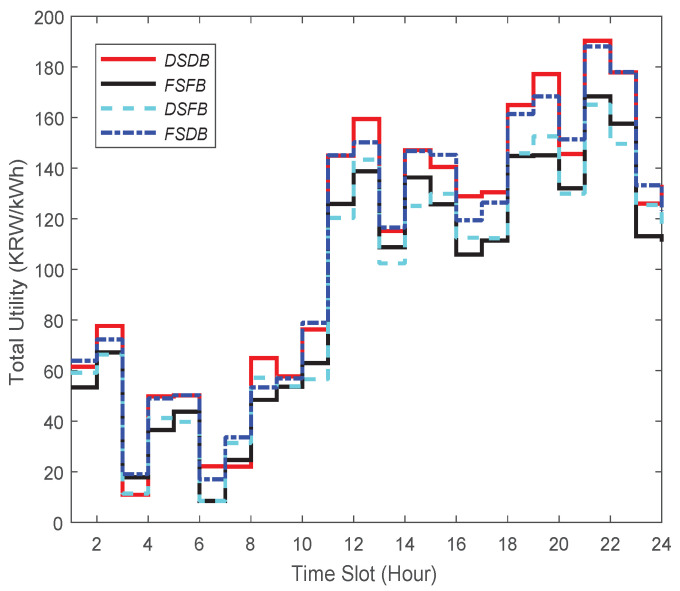
Total utility in the case in which the energy broker has high-cost ESS.

**Figure 9 sensors-21-05819-f009:**
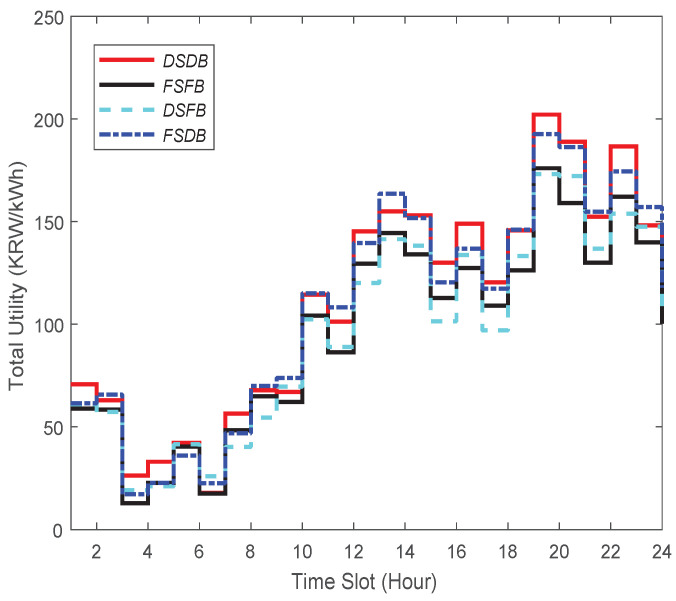
Total utility in the case in which the energy broker has low-cost ESS.

**Table 1 sensors-21-05819-t001:** Symbol notations.

Symbol	Description	Unit
$x_{i,k}$	the quantity of energy used by i-th consumer based on k-th unit time slot	kWh
$g_{i,k}$	the quantity of RES energy resold to the energy broker by i-th consumer in k-th unit time slot	kWh
$L_{i}$	the RES energy produced for the purpose of i-th consumer	kWh
$C_{i}$	the energy stored capacity of an ESS in i-th consumer	kWh
$c_{i,k}$	the quantity of energy preserved in k-th unit time slot in i-th consumer	kWh
$e_{i,k}$	the quantity of energy filled up with from the electric grid ($e_{i,k}$ ≥ 0) or discharged to the electric grid ($e_{i,k}$ ≤ 0) by i-th consumer in k-th unit time slot	kWh
$\delta_{i,k}$	the quantity of maximum charging of the ESS in k-th unit time slot in i-th consumer	kWh
$s_{i,k}$	the quantity of energy selling to energy broker in k-th unit time slot in i-th consumer	kWh
$R_{k}$	the quantity of RES energy that the energy provider purchases from consumers in the k-th unit time slot	REC/kg
*P*	the price value per unit carbon emissions (*P* > 0)	kW/kg
$M_{k}$	the quantity of carbon footprint in the k-th unit time slot	kg
$G_{k}$	the energy generated by the energy provider in k-th unit time slot	MW

**Table 2 sensors-21-05819-t002:** Total utility comparison under present and future ESS deployment cost.

	DSDB	FSFB	DSFB	FSDB
Total Utility (cent) for High ESS’s Cost	221.3	192.6	194.0	218.9
Total Utility (cent) for Low ESS’s Cost	229.1	200.1	201.1	223.3

## Data Availability

Not applicable.

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
