# Peer review of "Energy Trading among Power Grid and Renewable Energy Sources: A Dynamic Pricing and Demand Scheme for Profit Maximization"

_sensors, 2021, doi:10.3390/s21175819_

Round 1

Reviewer 1 Report

The paper addresses an important topic. The methods are sound and it provides valuable findings. Conclusions can be expanded to summarize numerical results and better illustrate possible impact and applications.

Reviewer 2 Report

The article deals with very important and current problems regarding the optimization of energy use in terms of generation of energy from renewable sources. The mathematical approach to the optimization problem is presented, assuming dynamically changing energy prices. In my opinion, the problem is purely theoretical. In fact, there are several other aspects to consider, namely the actual grid layout, power transmission capacity, energy losses, etc.
However, I believe the article is worth publishing. Maybe the authors will consider extending the calculations and presenting other examples? The ESS is not presented in detail. There is no information about the actual capacity of the accumulator batteries.

Reviewer 3 Report

In my opinion, the paper is in general interesting and nice to read. The manuscript deserves to be published only once the authors fix the following issues.

General

  1. The addressed problem could be solved alternatively by a centralized framework. the authors should fully justify the need of a multi-agent framework? The motivation could be satisfying privacy or reduce computational and communication effort. In both case the case study should be enhanced with discussion and finding about if the aimed goal of the multi-agent framework are fully accomplished.

Literature review

  1. The main contributions of the paper are clearly described. Nevertheless, from the current manuscript it is not grasp understanding the novelty of the work. The authors should better highlight the innovative aspects of their work in the manuscript.

System design

  1. All the used variable in all the formula and figures should report the unit.

Problem formulation

  1. The authors should clearly characterize the overall problem that they intend to solve. What type of decision variables (i.e. integer, real, etc) and how many? How many constraints (bounding, inequality, equality)?
  2. The proposed model relies on the prediction of load demand and production curve. Several recent scientific studies on energy scheduling/trading show that the integration of forecast techniques improve the performance of energy management. The Authors should comment this point.     (a).  P. Scarabaggio et al., "Distributed Demand Side Management With Stochastic Wind Power Forecasting," in IEEE Transactions on Control Systems Technology, doi: 10.1109/TCST.2021.3056751.       (b).  Nassourou, M. et al. Robust Economic Model Predictive Control Based on a Zonotope and Local Feedback Controller for Energy Dispatch in Smart-Grids Considering Demand Uncertainty. Energies 2020, 13, 696.

(documents that could be cited in the text)

  1. The authors should clarify how they handle the uncertainty of parameters. Several recent scientific studies on power grid, show that robust optimization instead of deterministic is a viable technique to deal with uncertainty of parameters. The Authors should comment this point.      (a).  Karimi, H., and Jadid, S. (2020). Optimal energy management for multi-microgrid considering demand response programs: A stochastic multi-objective framework. Energy, 195, 116992.         (b).  Melhem et al., "Energy Management in Electrical Smart Grid Environment Using Robust Optimization Algorithm," in IEEE Transactions on Industry Applications, vol. 54, no. 3, pp. 2714-2726, 2018.

(documents that could be cited in the text)

Case study

  1. Where data come from?
  2. No scalability analysis is provided in the case study, even though this could justify the proposed multi-agent framework.

Conclusions

  1. Conclusions needs to be extended to present further implications for future research and many managerial insights based on the results of the study, as well as limitations.

Minor

  1. The authors should check that all the used acronyms are explained. 
  2. Mainly the English is good and there are only a few typos. However the paper should be carefully rechecked.

Round 2

Reviewer 3 Report

Previous comments and concerns have been sufficiently addressed. In the revised paper several improvements have been added.